# A Systematic Review and Meta-Analysis of the Efficacy of Evening Primrose Oil for Mastalgia Treatment

**DOI:** 10.3390/ijerph18126295

**Published:** 2021-06-10

**Authors:** Lina Liana Ahmad Adni, Mohd Noor Norhayati, Ritzzaleena Rosli Mohd Rosli, Juliawati Muhammad

**Affiliations:** Department of Family Medicine, School of Medical Sciences, Health Campus, Universiti Sains Malaysia, Kubang Kerian 16150, Kelantan, Malaysia; drlinaliana@student.usm.my (L.L.A.A.); ritzz.rosli@student.usm.my (R.R.M.R.); juliawati@usm.my (J.M.)

**Keywords:** vitamin E, danazol, NSAIDS, breast pain

## Abstract

Mastalgia, or breast pain, is common among women which can lead to significant impairment in daily living. Hence, finding an effective treatment that can alleviate the symptom is very important. Thus, we carry out this study to determine the efficacy of evening primrose oil (EPO) for mastalgia treatment in women. The review included published randomised clinical trials that evaluated EPO used for treating mastalgia against a placebo or other treatments, irrespective of the blinding procedure, publication status, or sample size. Two independent authors screened the titles and abstracts of the identified trials; full texts of relevant trials were evaluated for eligibility. Two reviewers independently extracted data on the methods, interventions, outcomes, and risk of bias. The random-effects model was used for estimating the risk ratios and mean differences with 95% confidence intervals. Thirteen trials with 1752 randomised patients were included. The results showed that EPO has no difference to reduce breast pain compared to topical NSAIDS, danazol, or vitamin E. The number of patients who achieved pain relief was no different compared to the placebo or other treatments. The EPO does not increase adverse events, such as nausea, abdominal bloating, headache or giddiness, increase weight gain, and altered taste compared to a placebo or other treatments. EPO is a safe medication with similar efficacy for pain control in women with mastalgia compared to a placebo, topical NSAIDS, danazol, or vitamin E.

## 1. Introduction

Breast pain is a common cause of anxiety amongst women and frequently leads to primary care clinic consultation [1]. It is also the most common symptom amongst patients visiting the breast clinic [2]. Approximately 60 to 70% of women experience some type of breast pain, and in 10 to 20% of the cases, it is severe [3,4]. The pain has been more commonly reported amongst older women, those with larger breast sizes, and those less fit. Besides that, of those reported symptoms, 48 and 37% reported a negative impact from breast pain on their sexual activity and physical function, respectively. Ten percent of those symptomatic had reported breast pain that interfered with their social functioning [5]. Another alarming concern of patients presenting with mastalgia is the fear of breast cancer.

Mastalgia can be classified into cyclical mastalgia, non-cyclical mastalgia, and extramammary pain. Clinically, it is more important to differentiate between extramammary and true breast pain than cyclic and non-cyclic. It is because the management of cyclic and non-cyclic is similar, whilst extramammary pain may require a different treatment. Cyclical mastalgia affects two-thirds of patients with true mastalgia. Cyclical mastalgia occurs one to two weeks before menses and is relieved by the onset of the menstrual flow. It may be caused by normal cyclical changes in hormones [5]. It is also often described as premenstrual mastodynia and usually responds very well to hormonal therapies [6]. The pain is commonly diffuse and bilateral, with some radiation to the upper arm and axilla. These patients are usually aged between their 30s and 40s. Cyclical mastalgia may have a spontaneous resolution in up to 22% of the patients and persists in up to 65% of the patients after treatment [7].

Non-cyclical mastalgia affects one-third of the patients, and it is characterised by intermittent or constant breast pain and does not seem to be associated with the menstrual cycle. The etiology of non-cyclical mastalgia is usually unclear and appears to less likely to respond to hormonal therapies [8]. It is usually unilateral and localised to a particular quadrant of the breast. Patients are usually older, in their 40s and 50s, and are often perimenopausal [9]. Non-cyclical mastalgia can resolve without treatment in up to 50% of the cases but can also be more challenging to treat [10].

Extramammary pain is linked to various clinical disorders, such as costochondritis, Tietze syndrome, arthritis, slipping, and clicking ribs [11]. There are three leading causes of breast pain, which are an increase in estrogen or prolactin and a decrease in progesterone [7]. Works of literature have shown that patients with mastalgia had a significantly higher rise in prolactin [12].

The seeds of the evening primrose are rich in omega-6 essential fatty acids (EFAs), including linoleic acid and gamma-linolenic acid (GLA) [13]. Women with breast pain have low levels of GLA and its metabolite. Thus, treatment with EPO will raise the levels of GLA and its metabolites towards normal and probably relieves breast pain [14]. The therapeutic effects of EPO are attributed to the direct action of its component EFAs on immune cells as well as their indirect effect on the synthesis of eicosanoids (e.g., prostaglandins, cytokines, cytokine mediators), which are significantly high in mastalgia [13].

The first-line therapy for mastalgia is usually conservative, which involves physical support, over the counter analgesics, and manipulation of hormone-based medication for at least six months. If not responded to, treatment will be upgraded to the second-line therapy, such as tamoxifen and danazol, which may be more effective but not without adverse effects. The EPO is more compatible with the human body and has fewer side effects because it is made from a natural substance. The most common reported adverse effects include gastrointestinal upset (e.g., abdominal pain, indigestion, nausea, softening of stools) and headaches [15,16]. Supplementation of EPO will possibly improve the wellbeing and overall quality of life of a woman with mastalgia.

## 2. Materials and Methods

Our systematic review was conducted according to the protocol previously published in the PROSPERO register. The methodology and reporting were based on recommendations from the Cochrane Collaboration [17] and the preferred reporting items for systematic reviews and meta-analyses statement that the evaluation was conducted according to the Grading of Recommendations Assessment, Development, and Evaluation (GRADE) guidelines [18].

### 2.1. Eligibility Criteria

We considered randomised clinical trials (RCTs) comparing EPO with either a placebo or other drugs that are commonly used in treating mastalgia. We included blinded and open-label studies. The interventions were oral EPO, which should be taken every day. The comparisons included a placebo or other drugs commonly used for treating mastalgia. There was no restriction on age or other comorbidities. Publications that were published in other languages were translated into English.

### 2.2. Search Strategy

We searched the Cochrane Central Register of Controlled Trials (CENTRAL 2021, Issue 4), MEDLINE (1946 to February 2021), EMBASE, and Epistomonikus. We used the text words ‘evening primrose oil’, ‘gamolenic acid’, ‘mastalgia’, and ‘breast tenderness’, and Boolean operators, such as AND, OR, truncation, and wildcards for a variation in words. We checked the references list of the identified RCTs and reviewed articles to find unpublished trials or trials not identified by the electronic searches. We also searched for ongoing trials through the World Health Organisation (WHO), International Clinical Trials Registry Platform (ICTRP), and ClinicalTrials.gov (8 February 2021).

### 2.3. Trial Selection

Two review authors scanned the titles and abstracts independently from the searches and obtained full-text articles when they appeared to meet the eligibility criteria, or there was insufficient information to assess eligibility. We independently assessed the eligibility of the trials and documented the reasons for exclusion. We resolved any disagreements between the review authors by discussion. We contacted the trial authors if clarification was needed.

### 2.4. Data Extraction

The review authors (L.L.A.A., N.M.N.) independently extracted the characteristics of the trials (study setting), participants’ characteristics (age, sex, ethnicity, and comorbidities), methodology (number of participants randomized and analyzed, duration of treatment, and follow up), description of the intervention (dosage and route of administration), and outcomes into data extraction forms. When information was missing or inadequately reported, we contacted the corresponding authors for the trial.

The primary outcomes included the severity of pain reported as a mean pain score and the number of patients who responded to treatment or achieved clinical responses with EPO. The secondary outcomes were the occurrences of adverse events and quality of life.

### 2.5. Assessment of Risk of Bias

We assessed the risk of bias based on random sequence generation, allocation concealment, blinding of participants and personnel, blinding of outcome assessors, completeness of outcome data, the selectivity of outcome reporting, and other bias, as discussed in the Cochrane Handbook for Systematic Reviews of Interventions [17]. We categorised the risk of bias into low, unclear, or high.

### 2.6. Statistical Analysis

All the statistical analyses were performed using Review Manager (RevMan) version 5.3.5 (Nordic Cochrane Centre, Cochrane Collaboration). For all the included trials with categorical outcomes, we calculated the risk ratio (RR) and 95% confidence interval (CI). For the numerical outcomes, we calculated the mean difference (MD) and 95% CI. If data from two or more trials were included in an analysis of an outcome, we reported the results of the random-effects model. We pooled these measures into meta-analyses and drew forest plots.

We checked the included trials for the unit of analysis errors. If we had encountered any cluster RCTs, we intended to adjust the results from the trials showing a unit of analysis error based on the mean cluster size and intra-cluster correlation coefficient [17].

We assessed the presence of heterogeneity via two steps. First, we assessed evident heterogeneity at face value by comparing populations, settings, interventions, and outcomes. Second, we assessed statistical heterogeneity utilising the I^2^ statistic [17]. Thresholds for the interpretation of the I^2^ statistic can be misleading since the importance of inconsistency depends on several factors. We used the following guide to interpret the heterogeneity: 0–40% may not be important, 30–60% may represent moderate heterogeneity, 50–90% may represent substantial heterogeneity, and 75–100%, represents considerable heterogeneity [17]. We explored the potential sources of heterogeneity when significant heterogeneity was present. We performed a sensitivity analysis to investigate the impact of the risk of bias for sequence generation and allocation concealment of the included studies.

### 2.7. Grading Quality of Evidence

We used the principles of the GRADE approach for evaluating the quality of evidence in the systematic reviews [18]. This approach specifies four levels of quality, the highest of which is for randomised trial evidence. It can be downgraded to moderate, low, or even very low-quality evidence, depending on the presence of the following four factors: (i) limitations in the design and implementation of available studies, (ii) indirectness of evidence, (iii) unexplained heterogeneity or inconsistency of results, and (iv) imprecision of results. We used the GRADEpro GDT software (Evidence Prime, Inc., Hamilton, ON, Canada) for reflecting the quality of the evidence for each outcome, and the assessment was phased in together with the summary of findings (SoF) table.

### 2.8. Patient and Public Involvement

We did not involve patients in the development of the research question, development of the study outcomes, and design of the study, or the conduct of the study.

## 3. Results

### 3.1. Trial Selection

We retrieved 72 records. We screened a total of 49 records. Following this, we reviewed the full texts of 17 studies (Figure 1). We identified 13 articles that met the review inclusion criteria and reported the outcomes. Four trials were excluded because they were clinical experiences and not randomised clinical trials [19,20,21,22]. There was no ongoing trial found during the search process.

### 3.2. Characteristics of the Trials

We included 13 trials with a total of 17 participants. The 12 trials contributed to the primary outcomes, which were the severity of pain reported as the mean pain score and the total number of patients responding to the treatment or patients achieving clinical responses. Seven trials reported the secondary outcomes, which were the occurrence of adverse effects of EPO and quality of life [23,24,25,26,27,28,29]. Table 1 summarises the characteristics of the included trials.

### 3.3. Participants

Four of the nine trials were conducted in high-income and developed countries in Europe [6,25,26,28], five trials were conducted in Iran [24,30,31,32,33], two trials were conducted in Pakistan [23,27], and one trial was conducted in India [34] and Iraq [29], respectively. Eleven trials were single-centre studies, and one trial was a multicentre study [28]. The trials were conducted in patients that had been diagnosed with mastalgia. In all the trials, patients were excluded if they had a history of taking EPO at least three months before the trials.

### 3.4. Interventions

The primary drug that was investigated in this review was EPO, where its efficacy was compared to a placebo, topical NSAIDS, danazol, or vitamin E. The route of administration for the EPO in all trials was oral. The range of doses of EPO given per day was 1–4 g. The range of the doses of the vitamin E doses given in four trials [24,30,31,33] ranged from 400–1200 IU per day, and the doses of danazol given in two trials [27,34] were 200 mg per day. The EPO was given for two months [24,30,33,34], three months [23,26,27,32], six months [6,25,31], or 12 months [28]. The patients were monitored during enrollment or on admission day [6,25,26,28,31,32,33], at three months [25,26,33], and six months [6,25,31] to assess the severity of pain.

The number of patients responding to treatment were monitored on admission day [23,26,27,34], at one month [23,27,30,34], two months [23,33,34], or three months [23,26,27,32]. The occurrences of the adverse effects were monitored after one month [23,24,27], two months [23,24,25], three months [23,25,27], and six months [25].

Five trials were compared against a placebo [6,25,26,28,32]; one trial compared EPO with 20 mg of a starch tablet per day for three months [26]. One trial compared EPO with 6 g of corn oil in a day for six months [6]. Another trial compared EPO with 3 g of corn oil and wheat-germ oil for six months [25]. One trial compared EPO with coconut oil for four months [28].

Four trials compared EPO with vitamin E [24,30,31,33]. The doses of vitamin E were given at 400 IU per day at two and six months in two trials [24,31,33] and 1800 mg in a day for two months in one trial [30]. Two trials compared EPO with danazol [27,34]. The doses of danazol given were 200 mg in a day for three months [27] and two months [34]. Two trials compared EPO with topical NSAIDS. The first trial used a Piroxicam gel (Feldene) 0.5% local application for three months [23], whilst another trial used Diclofenac Sodium (Olfen) 50 g gel for three months duration [29].

### 3.5. Outcomes

This review included two-arm [23,27,29,30,34], three-arm [24,32,33], and four-arm studies [6,25,26,28,31], where EPO was investigated with a placebo and other treatment options. All trials measuring the primary outcomes were included in the meta-analysis. Five trials reported the results of EPO versus placebo or no treatment [6,25,26,28,32]. Four trials involved comparisons with vitamin E [24,30,31,33], two with danazol [27,34], and two with topical NSAIDS [23,29].

The first primary outcome, i.e., the severity of pain reported as the mean pain score, was reported in seven trials [6,25,26,28,31,32,33]. The severity of pain was measured by using the McGill pain questionnaire in four trials [6,26,28,31]. Another trial defined the severity of pain as the sum of pain scores during three menstrual cycles divided by the recorded number of days with pain in that same period [25]. The change of the pain severity was defined as the difference in severity between the second half of the treatment period and the run-in period [25].

The second primary outcomes were the number of patients responding to treatment or the number of cases achieving clinical responses, which were measured by using the Cardiff Breast pain Score in six trials [23,26,27,28,30,34]. Grade 1 and 2 were evaluated as ‘effective response to treatment’. Grade 3 and 4 were considered as not responding to treatment.

The secondary outcomes, i.e., the occurrences of adverse effects, were measured in six trials [23,24,25,27,28,29]. The reported adverse effects were nausea [23,24,25,27,28,29], abdominal bloatedness or fullness [23,25,27], headache and giddiness [23,24,28], increased body weight [23,25], and altered or bad taste [23,27,29].

The quality of life assessment was measured in only one trial [26]. It was derived from the Breast Pain Questionnaire, which was composed of four individual scores: (i) the sum of sensory and affective scores obtained from each sensory and affective descriptor, (ii) the present pain index calculated as a percentage, (iii) the score obtained from a visual analog scale, and (iv) the score for quality of life questions.

### 3.6. Assessment of the Risk of Bias

The assessment of the risk of bias is shown in Figure 2 and Figure 3. Figure 2 shows the proportion of studies assessed as having low, high, or unclear risks of bias for each risk of bias. Figure 3 shows the risk of bias for individual studies.

#### 3.6.1. Allocation

All trials described the method of randomisation for participant allocation except for three trials [26,29,33]. Two trials applied the simple randomisation technique [30,31], one trial used block randomisation [25], and one trial used a randomisation table [6]. Two trials applied a quasi randomisation technique [24,34]. Ref. [34] and the other three trials used a simple randomisation technique [23,27,29,32]. Allocation concealment was not mentioned in eight trials [24,25,26,27,31,32,33,34] and concealment was not performed in one trial [23]. Only one trial mentioned their method of concealment, where concealment was packaged, labelled, and distributed by the research pharmacy [6].

#### 3.6.2. Blinding

Eight trials did not mention the blinding of participants [23,24,26,27,30,32,33,34], while both participants and personnel were blinded in four trials [6,25,28,31]. All trials did not mention the blinding of the outcome assessment.

#### 3.6.3. Incomplete Outcome Data

Five trials [6,24,25,26,30] had a loss to follow-up cases, but it was unlikely to be related to the outcome, and another six [23,27,31,32,33,34] trials had no details of withdrawal or loss to follow up. One trial mentioned that they performed an intention-to-treat analysis [25]. The other trials did not mention the intention-to-treat principle analysis, but the participants were analysed according to the groups to which they were initially assigned [6,23,24,26,27,31,34].

#### 3.6.4. Selective Reporting

All 13 trials reported the outcomes as specified in their methods [6,23,24,25,26,27,28,29,30,31,32,33,34]. None of the trials were registered in the WHO ICTRP or ClinicalTrials.gov.

#### 3.6.5. Other Potential Sources of Bias

We detected no other potential sources of bias.

### 3.7. Clinical Outcomes

The primary outcomes were measured in 12 trials [6,23,25,26,27,28,29,30,31,32,33,34]. The secondary outcomes were measured in seven trials [23,24,25,27,28,29], which limited our analysis of the secondary outcomes to the occurrence of adverse effects and quality of life.

#### 3.7.1. Comparison between EPO and the Placebo

For the first primary outcome, the severity of pain reported as a mean pain score, there were five trials available [6,25,26,28,32]. The EPO did not reduce breast pain in comparison to taking the placebo (SMD −0.37, 95% CI −0.76 to 0.03, I^2^ = 73%, *p* = 0.070; five trials, 525 participants, high-quality of evidence) (Figure 4, Table 2). For the second primary outcome, there was one trial with data presented with the number of patients who responded to the treatment [26]. The EPO showed no clinical response (MD 1.26, 95% CI 0.86 to 1.84, *p* = 0.230; one trial, 63 participants, low-quality evidence).

The occurrence of adverse effects as a secondary outcome was reported in two trials [25,28].The EPO did not increase the occurrence of adverse effects (MD 0.89 95%, CI 0.65 to 1.23, I^2^ = 0%, *p =* 0.490, two trials, 325 participants, moderate-quality evidence) (Figure 5, Table 2). There was one trial that reported the quality of life of patients after treatment with EPO [26]. The EPO showed better quality of life (MD 4.00, 95% CI −6.52 to 14.52, *p =* 0.460; one trial, 63 participants, low-quality evidence).

#### 3.7.2. Comparison between EPO and Topical NSAIDs

For the primary outcome, there were two trials with data presented with the number of patients who responded to the treatment [23,29]. The EPO showed good clinical response (MD 0.60, 95% CI 0.43 to 0.84, I^2^ = 46%, *p* = 0.003; two trials, 120 participants, high-quality evidence) (Figure 6, Table 3).

The occurrence of adverse effects as a secondary outcome was reported in two trials [23,29]. The EPO did not show any difference from NSAIDs in the occurrence of adverse effects (MD 3.93, 95% CI 0.44 to 34.82, I^2^ = 0%, *p =* 0.220; two trials, 170 participants, high-quality evidence) (Figure 7, Table 3).

#### 3.7.3. Comparison between EPO and Danazol

For the primary outcome, there were two trials with data presented as the number of patients who responded to the treatment [27,34]. The EPO did not increase the total number of patients responding to the treatment or achieving clinical response (MD 0.71, 95% CI 0.36 to 1.40, I^2^ = 81%, *p =* 0.320; two trials, 175 participants, moderate-quality evidence) (Figure 8, Table 4).

The occurrence of adverse effects as a secondary outcome was reported in one trial [27]. The EPO did not increase the occurrence of adverse effects (MD 0.38, 95% CI 0.16 to 0.88, *p* = 0.020; one trial, 100 participants, low-quality evidence).

#### 3.7.4. Comparison between EPO and Vitamin E

For the first primary outcome, the severity of pain was reported as a mean pain score; three trials were available for this comparison [6,31,33]. The EPO did not reduce pain in comparison to Vitamin E (MD −0.47, 95% CI −1.07 to 0.14, I^2^ = 56%, *p* = 0.130; three trials, 305 participants, moderate-quality of evidence) (Figure 9, Table 5). For the second primary outcome, there was one trial with data presented with the number of patients who responded to the treatment [30]. The EPO showed good clinical response (MD 2.30, 95% CI 1.19 to 4.43, *p* = 0.010; one trial, 61 participants, low-quality evidence).

The occurrence of adverse effects as a secondary outcome was reported in one trial [27]. The EPO did not increase the occurrence of adverse effects (MD 3.21, 95% CI 0.14 to 75.61, *p* = 0.470; one trial, 58 participants, low-quality evidence).

## 4. Discussion

### 4.1. Summary of the Main Result

This review was designed to include all RCTs that focused on the effectiveness of EPO in treating women with mastalgia. The 13 identified trials addressed several comparisons of drugs and outcomes. This study shows that EPO has no difference in the reduction in the severity of pain in women with mastalgia compared to placebo, topical NSAIDS, danazol, or vitamin E. The number of patients who achieved pain relief was no different compared to the placebo or other treatments. EPO is a safe medication as it was not associated with the occurrence of adverse events, which were nausea, abdominal bloating, headache or giddiness, weight gain, and altered taste. The participants on EPO showed a better quality of life as compared to the control group; however, this outcome was derived from one trial, thus provides low-quality evidence.

### 4.2. Overall Completeness and Applicability of the Evidence

We performed a comprehensive and extensive literature review for assessing the effectiveness of EPO in treating mastalgia in a woman. The control groups ranged from placebo to vitamin E, danazol, and topical NSAIDS. The duration and doses of the EPO given to the patients were different in each trial, thereby limiting the applicability of the findings in this review. We were unable to do a subgroup analysis because of the limited number of trials available. We have been unable to determine the exact adverse effects of EPO due to the collective assessment of the adverse effects in all four trials included.

### 4.3. Quality of the Evidence

The quality of the evidence for the primary outcomes ranged from low to high. Although there were unclear and high risks of bias in some risk of bias assessments, we think that these risks were not significant for the review because pain itself is a subjective assessment, and the outcome would not have affected whether with health personnel, or participants, unblinded or not. We encountered high heterogeneity in the trials reporting response to the treatment and the occurrence of adverse events, which was probably due to the differences in the duration of the treatment or doses of the drugs. However, we were unable to perform a subgroup analysis for confirmation due to the limited number of available studies.

### 4.4. Potential Biases in the Review Process

We checked the reference lists of all related studies for further references and searched multiple databases. Articles that were not in the English language were translated. Despite the vigorous search of journal databases, we cannot be sure that we have extracted all trials relevant to our review.

### 4.5. Agreements and Disagreements with Other Studies or Reviews

The use of EPO did not offer a clear benefit or reduce pain in women with mastalgia or achieve a clinical response. This is consistent with one review, which reported that EPO did not cause a considerable decrease in mastalgia, and the EPO did not show any distressing adverse effects [35]. In another review, they also concluded that EPO, though commonly prescribed, is not effective [36]. One meta-analysis found that EPO did not offer any advantages over placebos in pain relief and should not be used [37].

## 5. Conclusions

### 5.1. Implications for Practice

Presently, the current guideline available for management of mastalgia includes first-line treatment, which consists of physical support (well-fitting bra or hot/cold compress), acetaminophen or NSAIDS (oral or topical), stop or reduce hormone replacement therapy or oral contraceptive, caffeine abstinence, and EPO [38]. In this review, it was found that EPO had no difference compared to the placebo or other treatment in reducing breast pain for women with mastalgia. The EPO does not increase adverse events, such as nausea, abdominal bloating, headache or giddiness, increase weight gain, and altered taste. The EPO is a safe medication with similar efficacy for pain control in women with mastalgia.

### 5.2. Implications for Research

More research is needed to determine the efficacy and safety of EPO for the treatment of mastalgia in women. Many of the previous trials did not mention the side effects or rate of withdrawal from the studies, and the reporting of adverse events following the treatment was not uniform throughout the studies. Thus, the meta-analysis of the trials will necessarily be incomplete due to a lack of data.

## Figures and Tables

**Figure 1 ijerph-18-06295-f001:**
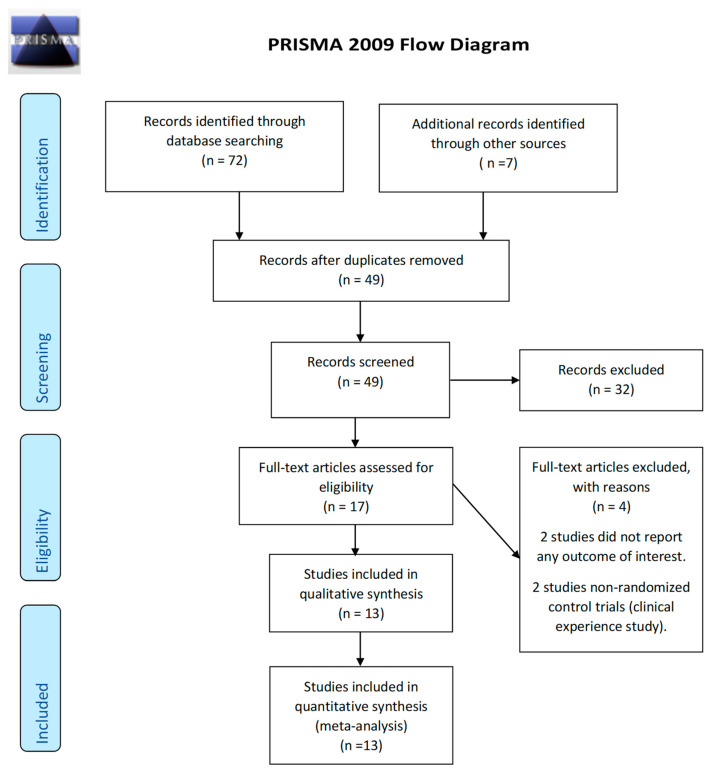
PRISMA study flow chart.

**Figure 2 ijerph-18-06295-f002:**
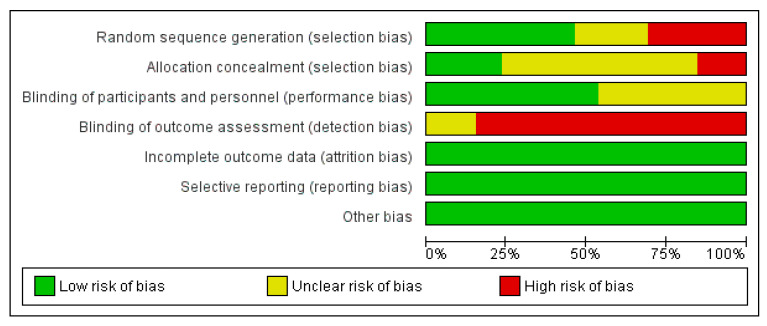
Judgement about each risk of bias item presented as percentages across all included studies.

**Figure 3 ijerph-18-06295-f003:**
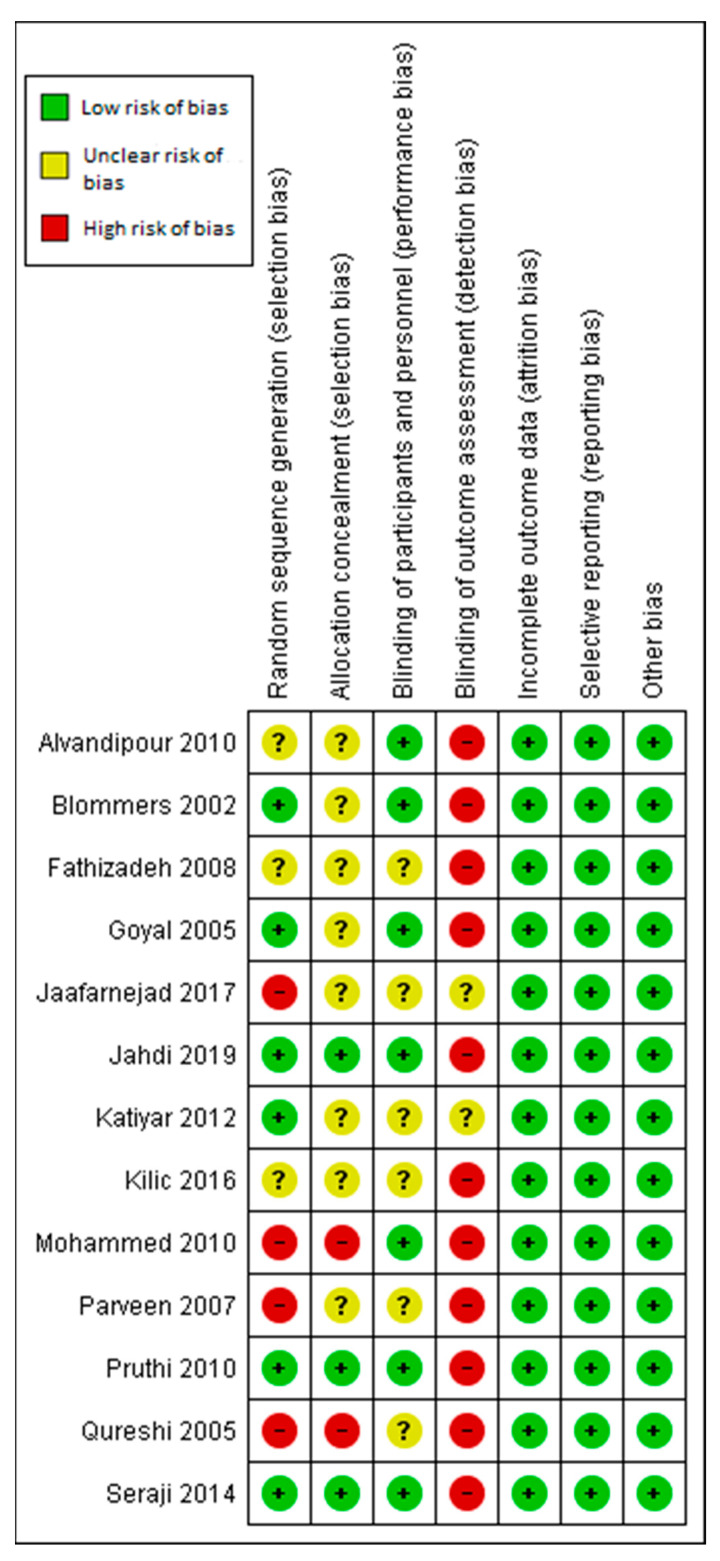
Judgements about each risk of bias item for each included study.

**Figure 4 ijerph-18-06295-f004:**
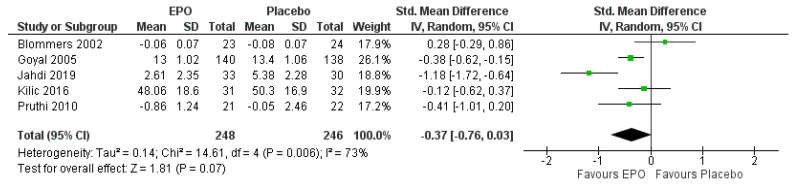
Forest plot of comparison between EPO and placebo for the outcome of the severity of pain.

**Figure 5 ijerph-18-06295-f005:**
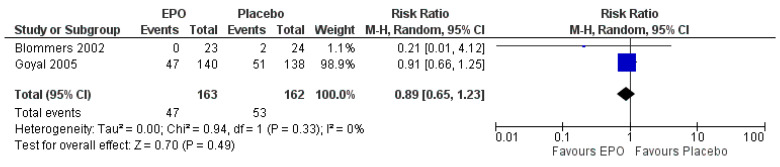
Forest plot of comparison between EPO and placebo for the outcome of the occurrence of adverse events.

**Figure 6 ijerph-18-06295-f006:**
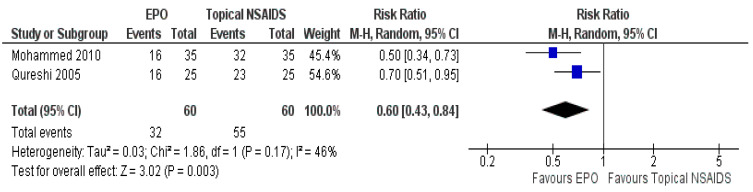
Forest plot of comparison between EPO and topical NSAIDs for the outcome of the number of patients responding to treatment.

**Figure 7 ijerph-18-06295-f007:**
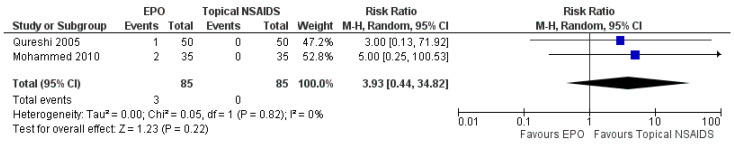
Forest plot of comparison between EPO versus topical NSAIDs for the outcome of the occurrence of adverse events.

**Figure 8 ijerph-18-06295-f008:**
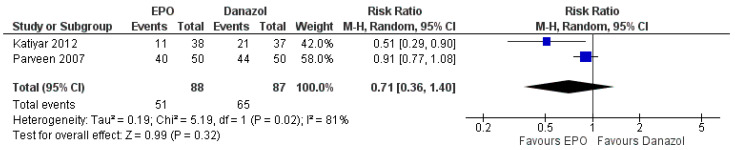
Forest plot of comparison between EPO and Danazol for the outcome of the number of patients responding to treatment.

**Figure 9 ijerph-18-06295-f009:**
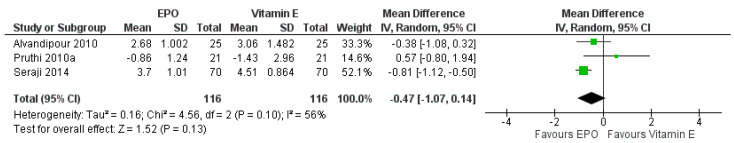
Forest plot of the comparison between EPO and vitamin E for the outcome of the severity of pain reported as a mean pain score.

**Table 1 ijerph-18-06295-t001:** Characteristics of included trial.

Studies	Setting	Size, n	Age	Evening Primrose Oil Dose (g)/Day	Comparison Dose/Day	Time of Treatment (Months)	Outcome
(1) Qureshi 2005	Karachi, Pakistan	50	15–50	1	Topical NSAIDs (Piroxicam gel 0.5%)	3	- The number of patients responding to treatment. - The occurrence of adverse events.
(2) Parveen 2007	Karachi, Pakistan	100	15–50	1	Danazol; 200 mg	3	- The number of patients responding to treatment. - The occurrence of adverse events.
(3) Jaafarnejad 2017	Mashad, Iran	90	18–45	1	Vitamin E; 400 IU	2	- The severity of pain reported as a mean pain score.
(4) Blommers 2002	Amsterdam, The Netherlands	120	18–45	3	Corn oil plus wheat-germ oil; 3 g	6	- The severity of pain reported as a mean pain score.
(5) Kilic 2016	Ankara, Turkey	128	18–60	1	Starch tablet; 20 mg	3	- The number of patients responding to treatment.
(6) Pruthi 2010	Minnesota, USA	85	19–56	3	Corn oil; 6 g	6	- The severity of pain reported as a mean pain score.
(7) Alvandipour 2017	Ghaemshar, Iran	100	18–50	2	Vitamin E; 400 IU	6	- The number of patients responding to treatment.
(8) Katiyar 2012	Kanpur, India	80	15–55	3	Danazol; 200 mg	2	- The number of patients responding to treatment.
(9) Fathizadeh 2008	Isfahan, Iran	66	18–40	3	Vitamin E; 1800 mg	2	- The number of patients responding to treatment.
(10) Goyal 2005	Cardiff, UK	555	18–55	4	Coconut oil; 500 mg	12	- The severity of pain reported as a mean pain score. - The occurrence of adverse events.
(11) Mohammed 2010	Baghdad, Iraq	70	17–48	3	Topical NSAIDs	3	- The number of patients responding to treatment - The occurrence of adverse events.
(12) Jahdi 2019	Tehran, Iran	94	18–50	2	Placebo capsule	3	- The severity of pain reported as a mean pain score.
(13) Seraji 2014	Arak, Iran	214	18–50	2	Vitamin E; 400 IU	2	- The severity of pain reported as a mean pain score.

**Table 2 ijerph-18-06295-t002:** Summary of the findings, including GRADE quality assessment for comparison between EPO and placebo.

EPO Compared to Placebo for Mastalgia Treatment
Patient or Population: Women with MastalgiaSetting: Outpatient ClinicIntervention: Evening Primrose OilComparison: Placebo
Outcomes	Anticipated Absolute Effects * (95% CI)	Relative Effect(95% CI)	No of Participants(Studies)	Certainty of the Evidence(Grade)	Comments
Risk with Placebo	Risk with EPO
The severity of pain	The mean severity of pain was 0	SMD 0.37 lower(0.76 lower to 0.03 higher)	-	525(5 RCTs)	⊕⊕⊕⊕HIGH	Risk of bias: not serious
Inconsistency: not serious
Indirectness: not serious
Imprecision: not serious

The number of patients responding to treatment	Study population	RR 1.26(0.86 to 1.84)	63(1 RCT)	⊕⊕⊝⊝LOW	Risk of bias: not serious
563 per 1000	709 per 1000(484 to 1000)	Inconsistency: not serious
Indirectness: not serious
Imprecision: not serious

The occurrence of adverse events	Study population	RR 0.89(0.65 to 1.23)	325(2 RCTs)	⊕⊕⊕⊝MODERATE	Risk of bias: not serious
327 per 1000	291 per 1000(213 to 402)	Inconsistency: not serious
Indirectness: serious
Imprecision:
not serious
Quality of life	The mean quality of life was 0	MD 4 higher(6.52 lower to 14.52 higher)	-	63(1 RCT)	⊕⊕⊝⊝LOW	Risk of bias: not serious
Inconsistency: serious
Indirectness: not serious
Imprecision:
serious

* The risk in the intervention group (and its 95% confidence interval) was based on the assumed risk in the comparison group and the relative effect of the intervention (and its 95% CI). CI: Confidence interval, RR: Risk ratio. GRADE Working Group grades of evidence. High certainty: We are very confident that the true effect lies close to that of the estimate of the effect. Moderate certainty: We are moderately confident in the effect estimate: The true effect is likely to be close to the estimate of the effect, but there is a possibility that it is substantially different. Low certainty: Our confidence in the effect estimate is limited: The true effect may be substantially different from the estimate of the effect. Very low certainty: We have very little confidence in the effect estimate: The true effect is likely to be substantially different from the estimate of effect.

**Table 3 ijerph-18-06295-t003:** Summary of the findings, including GRADE quality assessment for comparison between EPO and NSAIDS.

EPO Compared to Topical NSAIDs for Mastalgia Treatment						
Patient or Population: Women with Mastalgia Setting: Outpatient Clinic Intervention: Evening Primrose Oil Comparison: Topical NSAIDs						
Outcomes	Anticipated Absolute Effects * (95% CI)		Relative Effect (95% CI)	No of Participants (Studies)	Certainty of the Evidence (Grade)	Comments
	Risk with Topical NSAIDs	Risk with EPO				
The number of patients responding to treatment	Study population		RR 0.60 (0.43 to 0.84)	120 (2 RCTs)	⊕⊕⊕⊕ HIGH	Risk of bias: not serious
	917 per 1000	550 per 1000 (394 to 770)				Inconsistency: not serious
						Indirectness: not serious
						Imprecision: not serious
The occurrence of adverse events	Study population		RR 3.93 (0.44 to 34.82)	170 (2 RCTs)	⊕⊕⊕⊕ HIGH	Risk of bias: not serious
	0 per 1000	0 per 1000 (0 to 0)				Inconsistency: not serious
						Indirectness: not serious
						Imprecision: not serious

* The risk in the intervention group (and its 95% confidence interval) was based on the assumed risk in the comparison group and the relative effect of the intervention (and its 95% CI). CI: Confidence interval, RR: Risk ratio. GRADE Working Group grades of evidence. High certainty: We are very confident that the true effect lies close to that of the estimate of the effect. Moderate certainty: We are moderately confident in the effect estimate: The true effect is likely to be close to the estimate of the effect, but there is a possibility that it is substantially different. Low certainty: Our confidence in the effect estimate is limited: The true effect may be substantially different from the estimate of the effect. Very low certainty: We have very little confidence in the effect estimate: The true effect is likely to be substantially different from the estimate of the effect.

**Table 4 ijerph-18-06295-t004:** Summary of the findings, including GRADE quality assessment for comparison between EPO and danazol.

EPO Compared to Danazol for Mastalgia Treatment						
Patient or Population: Women with Mastalgia Setting: Outpatient Clinic Intervention: Evening Primrose Oil Comparison: Danazol						
Outcomes	Anticipated Absolute Effects * (95% CI)		Relative Effect (95% CI)	No of Participants (Studies)	Certainty of the Evidence (Grade)	Comments
	Risk with Danazol	Risk with Comparison EPO				
The number of patients responded to treatment	Study population		RR 0.71 (0.36 to 1.40)	175 (2 RCTs)	⊕⊕⊕⊝ MODERATE	Risk of bias: not serious
	747 per 1000	530 per 1000 (269 to 1000)				Inconsistency: serious
						Indirectness: not serious
						Imprecision: not serious
The occurrence of adverse events	Study population		RR 0.38 (0.16 to 0.88)	100 (1 RCT)	⊕⊕⊝⊝ LOW	Risk of bias: not serious
	320 per 1000	122 per 1000 (51 to 282)				Inconsistency: not serious
						Indirectness: not serious
						Imprecision: serious

* The risk in the intervention group (and its 95% confidence interval) was based on the assumed risk in the comparison group and the relative effect of the intervention (and its 95% CI). CI: Confidence interval, RR: Risk ratio. GRADE Working Group grades of evidence. High certainty: We are very confident that the true effect lies close to that of the estimate of the effect. Moderate certainty: We are moderately confident in the effect estimate: The true effect is likely to be close to the estimate of the effect, but there is a possibility that it is substantially different. Low certainty: Our confidence in the effect estimate is limited: The true effect may be substantially different from the estimate of the effect. Very low certainty: We have very little confidence in the effect estimate: The true effect is likely to be substantially different from the estimate of effect.

**Table 5 ijerph-18-06295-t005:** Summary of the findings, including GRADE quality assessment for comparison between EPO and vitamin E.

EPO Compared to Vitamin E for Mastalgia Treatment						
Patient or Population: Women with Mastalgia Setting: Outpatient Clinic Intervention: Evening Primrose Oil Comparison: Vitamin E						
Outcomes	Anticipated Absolute Effects * (95% CI)		Relative Effect (95% CI)	No Of Participants (Studies)	Certainty of the Evidence (Grade)	Comments
	Risk with Vitamin E	Risk with EPO				
The severity of pain	The mean severity of pain was 0	MD 0.47 lower (1.07 lower to 0.14 higher)	-	305 (3 RCTs)	⊕⊕⊕⊝ MODERATE	Risk of bias: not serious
						Inconsistency: not serious
						Indirectness: not serious
						Imprecision: serious
The number of patients responded to treatment	Study population		RR 2.30 (1.19 to 4.43)	61 (1 RCT)	⊕⊕⊝⊝ LOW	Risk of bias: not serious
	267 per 1000	613 per 1000 (317 to 1000)				Inconsistency: not serious
						Indirectness: not serious
						Imprecision: not serious
The occurrence of adverse events	Study population		RR 3.21 (0.14 to 75.61)	58 (1 RCT)	⊕⊕⊝⊝ LOW	Risk of bias: not serious
	0 per 1000	0 per 1000 (0 to 0)				Inconsistency: not serious
						Indirectness: not serious
						Imprecision: not serious

* The risk in the intervention group (and its 95% confidence interval) was based on the assumed risk in the comparison group and the relative effect of the intervention (and its 95% CI). CI: Confidence interval, RR: Risk ratio. GRADE Working Group grades of evidence. High certainty: We are very confident that the true effect lies close to that of the estimate of the effect. Moderate certainty: We are moderately confident in the effect estimate: The true effect is likely to be close to the estimate of the effect, but there is a possibility that it is substantially different. Low certainty: Our confidence in the effect estimate is limited: The true effect may be substantially different from the estimate of the effect. Very low certainty: We have very little confidence in the effect estimate: The true effect is likely to be substantially different from the estimate of effect.

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
