# Peer review of "A Systematic Review and Meta-Analysis of the Efficacy of Evening Primrose Oil for Mastalgia Treatment"

_ijerph, 2021, doi:10.3390/ijerph18126295_

Round 1
Reviewer 1 Report
This manuscript is clear, well-described, and overall is good manuscript. I have some minor comments:
1- in the cohorts described in this study, I do not see any attention to the age groups and/or other complications/health issues that might modulate response to EPO!
2- Where are the thirteen trials conducted? I think the authors can address Mastalgia frequency in specific population!
3- I am very hesitant to include the severity of pain in the criteria because this is variable and cant be controlled.
4- One of the limitation of this study is the variation of the EPO doses. One thing might be useful is to focus on one of the trials and try to build an association of EPO doses vs outcome.
Author Response
Point 1 : In the cohorts described in this study, I do not see any attention to the age groups and/or other complications/health issues that might modulate response to EPO!
Response 1: The age range in each study were similar; therefore, they were unlikely to influence the differences in the results, if any. The age distribution for each included study was mentioned in Table 1. Other complications or health issues was not mentioned across all 13 included studies. We have stated the inclusions and exclusions criteria for each study in Table 1.
|
Studies |
Age |
Inclusion criteria |
Exclusion criteria |
|
Qureshi 2005 |
15-50 |
- Patients with moderate to severe breast pain of two to three months duration. |
- Patients with discrete lump, nipple discharge, lactation, and obvious breast abscess. |
|
Parveen 2007 |
15-50 |
- Patients with moderate to severe breast pain. |
- Patients with discrete lumps, nipple discharge, lactation, pregnancy, and breast abscess. |
|
Jaafarnejad 2017 |
18-45 |
- Informed consent. - Having a cyclical breast pain for at least 5 consecutive days in two previous menstrual cycles. - Regular 21–35-day menstrual cycles. - Lack of any abnormal case in clinical breast examination. - No history of breast cancer or any breast masses in themselves or first-degree relatives such as mother and sister. - Absence of tumours and cancers in other organs. - No history of breast surgery and gel injections in the breast. - Lack of having constipation. |
- Pregnancy. - Taking hormonal drugs such as oral contraceptives. - Sensitivity or intolerance to supplements and medication side effects during the study. - Missed medication for three consecutive days or five periodic days. - Withdraw from the study.
|
|
Blommers 2002
|
18-45 |
- Cyclic or noncyclic mastalgia for >6 months. - An average of ≥7 and a minimum of 5 days with breast pain per menstrual cycle. - Premenopausal state. - At least one ovary in situ. - Informed consent. |
- Patients who had used fish oil, evening primrose oil, or another. Medication for mastalgia (danazol, tamoxifen, bromocriptine) during the last 3 months was excluded. - Women with nonbreast mastalgia, solitary breast lesions, menopausal complaints, other diseases, and pregnancy or pregnancy wish were also excluded |
|
Kilic 2016 |
18-60 |
The presence of breast pain for at least 3 months. |
- Inability to fill the questionnaire - Age under 18 years. - Having an organic breast disease such as cancer, abscess, infection and others. - Previous breast surgery. - Pregnancy, pregnancy wish and lactation. |
|
Pruthi 2010
|
19-56 |
- Premenopausal stage. - Age at least 18 years. - Cyclical mastalgia. - Failed conservative measures (e.g., use of a support bra, physician reassurance) after 1 month. - A score of 3 or greater on a breast-pain survey with pain scores from 1 to 10 (10 being the worst pain) was also required. - Participants age 40 years or older were required to have had a normal mammogram result. - Participants younger than 40 years, the focal area of pain was evaluated by targeted ultrasound examination or a mammogram.
|
- Pregnancy or lactation. Use vitamin E (>200 IU/day) or EPO in the previous two weeks. - Regular use of aspirin, nonsteroidal anti-inflammatory drugs, or anticoagulant therapy; use of danazol, bromocriptine or tamoxifen in the previous 3 months; and a prior diagnosis of breast cancer.
|
|
Alvandipour 2017
|
18-50 |
- Patient with cyclical breast pain. |
Not mentioned. |
|
Katiyar 2012 |
15-55 |
- Aged 15 years – 55 years with breast pain. - The pain had to be of such severity that the physical activity was curtailed. - Persistent marked pain throughout menstrual cycle. - The pain was not relieved by mild analgesic (NSAID) - The pain had to be episodic and exacerbated during the cycle. |
- Not taken hormonal therapy for the last 6 weeks. - Not pregnant and did not wish to become so within 6 months. - Undergone hysterectomy. |
|
Fathizadeh 2008
|
18-40 |
- Periodical breast pain and tenderness. - Having more than 3 months history of pain. - Having more than 3 days in each menstrual cycle. |
Not mentioned. |
|
Goyal 2005 |
18-55 |
- Normal menstrual cycles (average length 23-33 days), moderate to severe mastalgia of a minimum of 3-months duration requiring drug treatment, with at least 7 days of pain per menstrual cycle. |
- History of therapy with Efamast, evening primrose oil, bromocriptine, or danazol in the last 3 months - Breast pain not related to benign breast disease. |
|
Mohammed 2010 |
17-48 |
Moderate to severe mastalgia (breast pain that significantly affect patients’ social activities and sleep). |
Patients with discrete lump(s), nipple discharge and lactation with clinical evidence of mastitis. |
|
Jahdi 2019 |
18-50 |
- Moderate or severe mastalgia with no other malignant and benign breast diseases.
|
- Not taking medicines for reducing pain (such as danazol,tamoxifen,bromocriptine) over the past 3 months. - Not menopause. - Not pregnant. - Not breastfeeding - Not taking medicines for more than 5 days. - No psychological disease. - Unwillingness to continue taking medications as prescribed. |
|
Seraji 2014 |
18-50 |
Presence of breast pain for at least 3 months. |
- Not mentioned. |
Point 2: Where are the thirteen trials conducted? I think the authors can address Mastalgia frequency in specific population!
Response 2: We have added the “Setting” of the study in Table 1 as above. We have also stated that in the text in section 3.3.
|
Qureshi 2005
|
Outpatient Department of Surgery, Dow Medical College and Civil Hospital, Karachi, Pakistan. |
|
Parveen 2007
|
Out-patient Department of Surgical unit III (Ward-26) at Jinnah Postgraduate Medical Centre, Karachi.Pakistan. |
|
Jaafarnejad 2017
|
Mashhad University of Medical Sciences Gynaecology Clinic of Ghaem Medical Centre of Mashhad. Iran. |
|
Blommers 2002 |
Vrije Universiteit University Medical Centre. Amsterdam, The Netherlands. |
|
Kilic 2016 |
Department of General Surgery, School of Medicine, Turgut Ozal University, Ankara, Turkey.
|
|
Pruthi 2010
|
Mayo Clinic in Rochester, Minnesota and the University of Minnesota, Twin Cities. USA |
|
Alvandipour 2017
|
Surgical clinics, School of Medical Science Mazandaran University. Sari, Ghaemshar, Iran. |
|
Katiyar 2012 |
Outpatient Department of surgery, Rama Medical College & Hospital, Kanpur, India. |
|
Fathizadeh 2008
|
Isfahan health centre. Iran. Isfahan University of Medical Sciences. |
|
Goyal 2005 |
Department of Surgery, Wales College of Medicine, Cardiff University,United Kingdom. |
|
Mohammed 2010 |
Department of surgery clinic, Al Mustansiriya College of Medicine, Baghdad, Iraq. |
|
Jahdi 2019 |
Breast clinic of Milad hospital,Tehran,Iran. |
|
Seraji 2014 |
Department of general surgery, Arak University of Medical Sciences, Iran. |
Point 3: I am very hesitant to include the severity of pain in the criteria because this is variable and can’t be controlled.
Response 3: Agree. Pain is a difficult outcome to measure due to its subjective nature. However, we felt that it is important, similar to other possible subjective outcomes, such as quality of life, which is also based on subjective assessment. In this review, the pain severity assessment was measured using a standardised tool, i.e., e McGill pain questionnaire. It is a well-validated and multidimensional measure with extensive clinical research use. Patients rate their pain in sensory terms (e.g., sharp or stabbing) and affective terms (e.g., sickening or fearful), with 15 total descriptors. Each item is rated on a 4-point scale that ranges from none to severe. It also has a single visual analog scale (VAS) item for pain intensity and a visual rating scale (VRS) for rating the overall pain experience.
Point 4: One of the limitations of this study is the variation of the EPO doses. One thing might be useful is to focus on one of the trials and try to build an association of EPO doses vs outcome.
Response 4: Agree. We intended to perform a subgroup analysis (section 4.2) based on the dosage. However, due to the limited number of trials per subgroup, it was not feasible.

Reviewer 2 Report
Authors undertook a systematic and meta-analysis review to assess the efficacy of evening primrose oil (EPO) in treatment of mastalgia in women. Utilizing data curated from thirteen published randomized clinical trials on their evaluated research focus, they reported that while EPO had no increased risk of adverse events, it equally provided no comparative therapeutic advantage when compared to placebo, topical NSAIDs, Vitamin E or danazol in treatment of women with mastalgia.
COMMENTS:
Firstly and as alluded to by review content, EPO apparently provides no overall therapeutic benefit when used for mastalgia treatment in women, yet authors could not definitely state as such. Rather, authors ascribed that it reduced pain similar to other treatment methods, including placebo. Are authors suggesting that EPO usage is still relevant to mastalgia treatment, based on their review findings? Provide a clearer justification in manuscript.
Why does Table 1 include the summary information of only eleven clinical trial studies, and not 13? Moreover, the total population number currently presented in Table 1 totals 1444; which is far less than 1764 described in methods section. Authors need to provide complete information for their selected thirteen (13) clinical trial and re-draw Table 1, providing content about Jahdi et al 2019 & Seraji et al 2014.
For readers’ convenience, provide a figure legend indicating what the color code with symbols represent for Figure 3.
Consistency in use of abbreviation is needed, especially in abstract section wherein authors indicated EPO means evening primrose oil; yet kept using both intermittently within its text.
Author Response
Point 1: Firstly, and as alluded to by review content, EPO apparently provides no overall therapeutic benefit when used for mastalgia treatment in women, yet authors could not definitely state as such. Rather, authors described that it reduced pain similar to other treatment methods, including placebo. Are authors suggesting that EPO usage is still relevant to mastalgia treatment, based on their review findings? Provide a clearer justification in manuscript.
Response 1: we have rephrased our conclusion as below.
“The results showed that EPO has no difference in the reduction in the severity of pain compared to the placebo, topical NSAIDs, danazol, or vitamin E. The number of patients who achieved pain relief was no different compared to the placebo or other treatments. The EPO does not increase adverse events, such as nausea, abdominal bloating, headache or giddiness, increased weight gain, and altered taste compared to a placebo or other treatments. EPO is a safe medication with similar efficacy for pain control in women with mastalgia compared to a placebo, topical nsaids, danazol, or vitamin E.”
Point 2: Why does Table 1 include the summary information of only eleven clinical trial studies, and not 13? Moreover, the total population number currently presented in Table 1 totals 1444; which is far less than 1764 described in methods section. Authors need to provide complete information for their selected thirteen (13) clinical trial and re-draw Table 1, providing content about Jahdi et al 2019 & Seraji et al 2014.
Response 2: Thank you. We have corrected Table 1 as indicated.
Point 3: For readers’ convenience, provide a figure legend indicating what the color code with symbols represent for Figure 3.
Response 3: Thank you. We have added the figure legend as indicated.
Point 4: Consistency in use of abbreviation is needed, especially in abstract section wherein authors indicated EPO means evening primrose oil; yet kept using both intermittently within its text.
Response 4: We have maintained “evening primrose oil” in the abstract. For the text, we have indicated the abbreviation for evening primrose oil in its first use and used the abbreviation throughout the text.

Reviewer 3 Report
Authors reviewed the efficacy of EPO on mastalgia. Mastalgia is common symptom of women in the world.
- Please erase the authors name on methods part. Authors contributions are well described in the bottom of the manuscript.
- Please remove "Institutional Review Board Statement: In this section, you should add the Institutional Review 442 Board Statement and approval number, if relevant to your study. You might choose to exclude this 443 statement if the study did not require ethical approval. Please note that the Editorial Office might 444 ask you for further information. Please add “The study was conducted according to the guidelines 445 of the Declaration of Helsinki, and approved by the Institutional Review Board (or Ethics Commit-446 tee) of NAME OF INSTITUTE (protocol code XXX and date of approval).” OR “Ethical review and 447 approval were waived for this study, due to REASON (please provide a detailed justification).” OR 448 “Not applicable” for studies not involving humans or animals. 449
Informed Consent Statement: Please add “Informed consent was obtained from all subjects in-450 volved in the study.” OR “Patient consent was waived due to REASON (please provide a detailed 451 justification).” OR “Not applicable” for studies not involving humans. 452
Data Availability Statement: Please refer to suggested Data Availability Statements in section 453 “MDPI Research Data Policies” at https://www.mdpi.com/ethics." if they are not needed. - Please increase the quality of the figures. The resolution is low.
- In the discussion section, the subheading should be "4.1. Summary of the main result", etc.
- EPO showed similar effect on pain control with mastalgia compared to a placebo group. Also, one meta-analysis pointed that there is no advantage over placebos with EPO. Please discuss this fact in the discussion section. Please discuss the limitation of EPO in a balanced way.
Author Response
Point 1: Please erase the authors name on methods part. Author’s contributions are well described in the bottom of the manuscript.
Response 1: Thank you. We have done as suggested.
Point 2: Please remove "Institutional Review Board Statement: In this section, you should add the Institutional Review 442 Board Statement and approval number, if relevant to your study. You might choose to exclude this 443 statement if the study did not require ethical approval. Please note that the Editorial Office might 444 ask you for further information. Please add “The study was conducted according to the guidelines 445 of the Declaration of Helsinki, and approved by the Institutional Review Board (or Ethics Commit-446 tee) of NAME OF INSTITUTE (protocol code XXX and date of approval).” OR “Ethical review and 447 approval were waived for this study, due to REASON (please provide a detailed justification).” OR 448 “Not applicable” for studies not involving humans or animals. 449
Informed Consent Statement: Please add “Informed consent was obtained from all subjects in-450 volved in the study.” OR “Patient consent was waived due to REASON (please provide a detailed 451 justification).” OR “Not applicable” for studies not involving humans. 452
Data Availability Statement: Please refer to suggested Data Availability Statements in section 453 “MDPI Research Data Policies” at https://www.mdpi.com/ethics." if they are not needed.
Response 2 : Thank you. We have done as suggested.
Point 3: Please increase the quality of the figures. The resolution is low.
Response 3: We have changed all forest plot figures to PNG format. They are clearer now.
Point 4: In the discussion section, the subheading should be "4.1. Summary of the main result", etc.
Response 4: We have edited as suggested.
Point 5: EPO showed similar effect on pain control with mastalgia compared to a placebo group. Also, one meta-analysis pointed that there is no advantage over placebos with EPO. Please discuss this fact in the discussion section. Please discuss the limitation of EPO in a balanced way.
Response 5: We have edited the discussion in the “Summary of the main result” and “Agreements and disagreements with other studies or reviews”
“This study shows that EPO has no difference in the reduction in the severity of pain in women with mastalgia compared to placebo, NSAIDs, Danazol or vitamin E.”
“The use of EPO did not offer a clear benefit or reduce pain in women with mastalgia or achieve a clinical response. This is consistent with one review, which reported that EPO did not cause a considerable decrease in mastalgia, and the EPO did not show any distressing adverse effects [35]. In another review, they also concluded that EPO, though commonly prescribed, is not effective [36]. One meta-analysis found that EPO did not offer any advantages over placebos in pain relief and should not be used [37]".

Round 2
Reviewer 2 Report
Authors have made substantial improvements to the revised manuscript
Reviewer 3 Report
The revised manuscript is qualified to be published.